# Path Planning of USV Based on the Improved Differential Evolution Algorithm

Zhongming Xiao
*Navigation College*
*Dalian Maritime University*
Dalian, China
xiaozhongming@dlmu.edu.cn

Baoyi Hou
*Navigation College*
*Dalian Maritime University*
Dalian, China
houbaoyi@dlmu.edu.cn

Jun Ning
*Navigation College*
*Dalian Maritime University*
Dalian, China
junning@dlmu.edu.cn

Bin Lin
*Information Science and Technology College*
*Dalian Maritime University*
Dalian, China
binlin@dlmu.edu.cn

Zhengjiang Liu
*Navigation College*
*Dalian Maritime University*
Dalian, China
liuzhengjiang@dlmu.edu.cn

*Abstract*—**Planning a reasonable path and avoiding collisions with surrounding obstacles are among the most critical aspects of Unmanned Surface Vehicle (USV) navigation, which has drawn considerable attention from researchers in recent years, with various heuristic and intelligent optimization algorithms being applied to path planning. However, most existing algorithms have not sufficiently integrated safety and economy, leading to the planned paths that may not align with maritime practice. Therefore, to tackle the aforementioned issues, this paper introduces a differential evolution algorithm (DE) with an adaptive crossover factor for path planning and collision avoidance in USV. The collision risk index (CRI) is integrated with the DE, and the CRI is improved by introducing a restriction factor when selecting the degree of membership for the distance to closest point of approach (DCPA). The experimental results demonstrate that, compared with the other three algorithms, the improved DE exhibits greater advantages in terms of minimum distance to the target ship, minimum distance to obstacles, and total yaw distance, thereby validating the effectiveness of the algorithm.**

*Index Terms*—**path planning, collision avoidance, collision risk index, differential evolution algorithm.**

## I. Introduction

Unmanned surface vehicles (USVs) are intelligent control system that integrates path planning, communications, autonomous decision-making, and automatic target recognition, as well as a range of other advanced technologies. USVs utilize radar and AIS to continuously monitor their surroundings, enabling dynamic adjustments in course and speed to effectively avoid collisions with other ships or unknown obstacles at sea. With the continuous development of USV technology, the operational capabilities of USVs in various complex marine environments have steadily improved. Consequently, USVs are being increasingly used in diverse domains of daily life, such as waterway patrol and safety monitoring, ocean exploration and geological surveys, and marine biodiversity conservation.

Path planning and collision avoidance technologies, as the core technologies of USV, have played a crucial role in their development. In light of this, scholars have conducted extensive research on the technologies. In past studies, many researchers have applied various heuristic algorithms to USV path planning, such as the A* algorithm[1] and the Dijkstra algorithm. With continuous development, many intelligent optimization algorithms have gradually been applied to the problem of path planning, such as the Ant Colony Optimization (ACO)[2] algorithm, Particle Swarm Optimization (PSO), Genetic Algorithm (GA)[3], Rapidly-exploring Random Tree (RRT) algorithm[4], Velocity Obstacle method (VO)[5], and Dynamic Window Approach (DWA). These algorithms derive feasible paths through specific operational strategies. However, during path planning, they often encounter issues such as falling into local optima or planning paths that are too close to obstacles, resulting in suboptimal solutions. Therefore, many researchers have improved various algorithms, such as the improved RRT algorithm[6], which introduces adaptive step size and target attraction mechanisms, allowing the USV to adaptively adjust its step size based on different waters and to adjust its direction of movement accordingly. The improved DWA[7] introduces the concept of obstacle search angle, enhancing the USV's obstacle avoidance capabilities in different scenarios.

To fully utilize the advantages of various algorithms, scholars have combined different algorithms. For example, the combination of the PSO and Artificial Potential Field (APF) method[8] first plans a global path using the improved PSO, and the improved APF method is used for local path planning when dynamic obstacles are detected during navigation, which effectively reduces the collision risk. The combination of the GA and the ACO[9] uses the solution from the ACO as the initial population for the GA, thereby accelerating the convergence speed. However, most existing algorithms have not

The work was supported by the National Natural Science Foundation of China (No. 51939001, No. 62371085) and Fundamental Research Funds for the Central Universities (No.3132023514).

*Corresponding author: Jun Ning.

sufficiently integrated safety and economy, leading to paths that may not align with maritime practice.

To address the various issues associated with the aforementioned algorithms, this paper proposes an Improved Differential Evolution algorithm (I-DE) and integrates it with the Collision Risk Index (CRI). Simulation experiments demonstrate that the I-DE, compared with the other three algorithms, can more effectively avoid collisions with target ships and obstacles while reducing deviation distance, ensuring both safety and economy. The primary contributions of this paper are outlined as follows:

(1) The crossover factor CR in the Differential Evolution algorithm (DE) is adaptively improved, enhancing population diversity while maintaining the relative independence of individuals. This allows the algorithm to search the solution space appropriately according to the different iteration stages.

(2) The CRI is integrated with the DE, and a restriction factor is added when selecting the degree of membership for the Distance to the Closest Point of Approach (DCPA). This makes the calculation of collision risk more aligned with maritime practice.

## II. System Model

### A. Differential evolution algorithm

Differential Evolution(DE)[10] is an algorithm used to solve continuous optimization problems. It primarily involves five steps: population initialization, fitness evaluation, differential mutation, crossover operation, and selection of new individuals.

*1) Population initialization:* Initially, a population of size M is formed by randomly generating M individuals, where each individual is composed of n-dimensional vector. The size of the population affects the search capabilities of the algorithm and the use of computational resources. Generally, a larger population enhances the algorithm's global search capabilities but also increases computational costs.

$$X_i(0) = (x_{i,1}(0), x_{i,2}(0), x_{i,3}(0),...,x_{i,n}(0)) \quad (1)$$

$$X_{i,j}(0) = X_{i\min} + rand(0,1)(X_{i\max} - X_{i\min}) \quad (2)$$

$$i = 1,2,3,...,M, j = 1,2,3,...n \quad (3)$$

Here, $X_i(0)$ denotes an individual, $x_{i,j}(0)$ denotes the j-th dimensional vector of the individual, with $X_{i\min}$ and $X_{i\max}$ specifying the respective lower and upper bounds of this vector.

*2) Fitness evaluation:* When calculating the fitness of the population individuals (the objective function value), it is necessary to define the objective function based on the specific problem. By designing appropriate objective functions, the algorithm can adapt to various optimization needs and complex problem environments, demonstrating high flexibility and adaptability. In this paper, the fitness is employed to assess the quality of the path points.

*3) Differential mutation:* Below are descriptions of several mutation strategies that have been extensively researched:
DE/rand/1:

$$V_i(G) = X_{r1}(G) + F \times (X_{r2}(G) - X_{r3}(G)) \quad (4)$$

DE/best/1:

$$V_i(G) = X_{best}(G) + F \times (X_{r1}(G) - X_{r2}(G)) \quad (5)$$

Using DE/rand/1 as an illustration, $X_{r1}(G)$, $X_{r2}(G)$, and $X_{r3}(G)$ are three different vectors randomly selected from the parent generation, $r1 \neq r2 \neq r3 \neq i \in \{1,2,3......,M\}$, F is the scaling factor, and F ranges from 0 to 2, typically set to 0.5. $V_i(G)$ is a new vector generated through the mutation strategy. Different mutation strategies have different population optimization abilities. To better understand the common properties of various mutation strategies, Feoktistov summarized them in a general form as follows: $V_i = \beta_i + F \times \delta_i$, where $\beta_i$ serves as the base vector and $\delta_i$ acts as the differential vector.

*4) Crossover operation:*

$$U_{i,j}(G) = \begin{cases} V_{i,j}(G), & rand[0,1) < CR \text{ or } j = jrand \\ X_{i,j}(G), & otherwise \end{cases} \quad (6)$$

The crossover factor CR ranges from 0 to 1. j is the current vector's dimension and $jrand$ is a dimension randomly selected within the range from 1 to n. Adding the condition $j = jrand$ guarantees that at least one dimension of the new individual comes from the mutant individual, thereby avoiding being identical to the initial individual. The crossover process is illustrated in Fig. 1.

*5) Selection of new individuals:* The selection operation evaluates the fitness values of individuals to steer the population toward a better direction. The direction of population evolution is determined by the following formula:

$$X_i(G+1) = \begin{cases} U_i(G), & f(U_i(G)) \leq f(X_i(G)) \\ X_i(G), & otherwise \end{cases} \quad (7)$$

Here, $f(U_i(G))$ and $f(X_i(G))$ are the fitness of the new individual and the initial individual, respectively.

### B. Ship encounter situations and responsibility allocation.

In areas with good visibility, collision avoidance behavior should comply with Rules 8, 13, 14, and 15 of the International Regulations for Preventing Collisions at Sea (COLREGs). Rule 8 explicitly stipulates the actions to be taken to avoid collisions, while Rules 13 to 15 define the different encounter situations: overtaking, head-on, and crossing encounters. Therefore, this paper incorporates the COLREGs and fully considers the implications of the ship encounter situations on collision avoidance behavior. Based on the course angles and positions of the two ships, the encounter between ships is classified into four scenarios. The classification is detailed in Table I.

When encountering another ship head-on, both ships share equal responsibility to give way. In overtaking situations, the overtaking ship has the responsibility to give way, while the ship being overtaken should preserve its original state. In a left-crossing scenario, the own ship should preserve the original state, with the other ship bearing the responsibility to give way. Conversely, in a right crossing situation, the own ship has the duty to give way, while the other ship should preserve the original state.

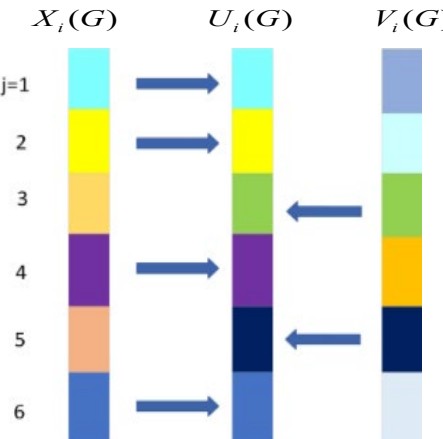

$$X_i(G) \quad U_i(G) \quad V_i(G)$$

Fig. 1 Crossover operation

TABLE I.
SHIP ENCOUNTER SITUATION CLASSIFICATION

| True bearing of TS to OS/° | Course difference/° | Encounter |
|---|---|---|
| $354 \le \theta_r \le 6$ | $174 \le \Delta C \le 186$ | Head-on |
| $247.5 \le \theta_r < 354$ | $67.5 \le \Delta C < 174$ | Left-Crossing |
| $6 < \theta_r \le 112.5$ | $186 < \Delta C \le 292.5$ | Right-Crossing |
| $112.5 < \theta_r < 247.5$ | $\Delta C < 67.5 \bigcup \Delta C > 292.5$ | Overtaking |

## III. ALGORITHM IMPROVEMENTS

This section introduces the I-DE. Firstly, the crossover factor CR in the crossover operation is adaptively improved[11]. Concurrently, enhancements are made to the traditional CRI model by incorporating a restriction factor when selecting the membership function for DCPA, thus aligning the calculation of CRI more closely with maritime practices. Finally, the COLREGs and CRI are incorporated into the fitness function evaluation, forming an evaluation set based on safety, economy, compliance with COLREGs, and optimal collision avoidance timing. The safety factor is determined by the CRI.

### A. Adaptive cross-factor CR

The crossover factor CR determines the likelihood of each dimension of an individual being altered. A larger CR value facilitates the more effective transfer of information from the mutant individual to the initial individual, while a smaller CR value, although reducing the transfer of information, enhances the independence between individuals. Therefore, an adaptive CR mechanism is proposed to balance the above two effects, with the following improvements:

$$C_{R_n} = \begin{cases} C_{R_1}, & f(x_n^G) > f(x_{avg}^G) \\ C_{R_0} \times \dfrac{(C_{R_1} - C_{R_0})(f(x_{avg}^G) - f(x_n^G))}{f(x_{avg}^G) - f(x_{min}^G)}, & f(x_n^G) \le f(x_{avg}^G) \end{cases} \quad (8)$$

$f(x_n^G)$ and $f(x_{avg}^G)$ denote the fitness of the n-th individual and the average fitness of all individuals, respectively. $f(x_{min}^G)$ denotes the lowest fitness across all individuals.

### B. Collision risk index (CRI)

The CRI[12] is a fuzzy index used to assess collision risk, representing the likelihood of a collision occurring between ships. It is affected by external factors like the speed and course of the ship, along with the subjective factors of the operator. This paper constructs a collision risk model using three factors: DCPA, TCPA, and the inter-ship distance D. Additionally, a restriction factor is added when selecting the membership function of DCPA to improve the rationality of the selection. The set of factors for the CRI is established as follows:

$$U = \{DCPA、TCPA、D\} \quad (9)$$

Define the membership functions for each factor:

#### 1) Membership function of DCPA

Take the own ship(OS)'s position as the origin to establish a spatial right-angled coordinate system, the OS's coordinates are set at $(x_O, y_O)$, and the speed and heading are set to $v_O$ and $\varphi_O$, respectively; similarly, the target ship(TS)'s position, speed and heading are set to $(x_T, y_T)$, $v_T$ and $\varphi_T$, respectively. The true bearings of OS to the TS and the TS to OS are $a_{OT}$ and $a_{TO}$, respectively. The relative speed between the two ships is $v_R$.

In previous studies, the selection of the membership function for DCPA only considered the safe distance of approach (SDA) $r_1$ and the absolute safe distance of approach $r_2$, without considering whether the ship domains of OS and the TS were infringed upon. Fig. 2 illustrates various situations where the ship domains of OS and the TS are infringed upon. Therefore, the membership function for DCPA is improved to address this issue.

Establish a coordinate system with the TS as the origin, the direction of the bow as the positive y-axis, and the direction perpendicular to the bow to the right as the positive x-axis. Perform a coordinate transformation for the position of OS:

$$x_{O1} = D\sin\beta_0, \, y_{O1} = D\cos\beta_0, \, \beta_0 = a_{OT} - \varphi_T + \gamma_1, \quad (10)$$

$$\gamma_1 = \begin{cases} 360, & a_{OT} - \varphi_T \le 0 \\ 0, & a_{OT} - \varphi_T > 0 \end{cases} \quad (11)$$

Based on the transformed coordinates $(x_{O1}, y_{O1})$, the relative motion line equation of OS relative to the TS is obtained:

$$y = \cot(\varphi_R - \varphi_T)x + (y_{O1} - x_{O1}\cot(\varphi_R - \varphi_T)) \quad (12)$$

$$\varphi_R = \begin{cases} \arctan\dfrac{v_{OTx}}{v_{OTy}} + \theta & otherwise \\ 90 & v_{OTx} \ge 0, v_{OTy} = 0 \\ 270 & v_{OTx} < 0, v_{OTy} = 0 \end{cases} \quad (13)$$

$$\theta = \begin{cases} 0 & v_{OTx} \ge 0, v_{OTy} > 0 \\ 180 & v_{OTx} \ge 0, v_{OTy} < 0 \text{ or } v_{OTx} < 0, v_{OTy} < 0 \\ 360 & v_{OTx} < 0, v_{OTy} > 0 \end{cases} \quad (14)$$

Due to the change in the coordinate system, the relative motion line equation also needs to be transformed:

$$x = \cot(\varphi_R - \varphi_T)y + (y_{O1} - x_{O1}\cot(\varphi_R - \varphi_T)) \quad (15)$$

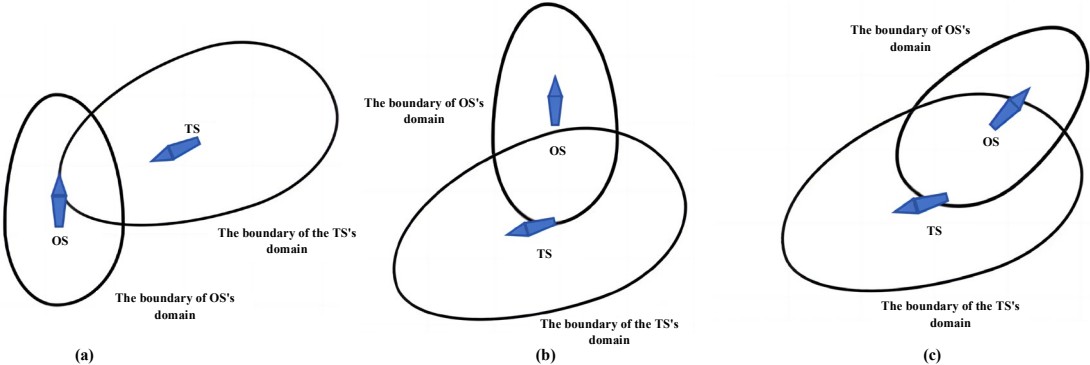

Fig. 2 Various situations of OS and the TS's domains being intruded upon: (a)the TS does not intrude into OS's domain, but OS intrudes into the TS's domain; (b)OS does not intrude into the TS's domain, but the TS intrudes into OS's domain; (c)both ships intrude into each other's domains.

As shown in Fig. 3, when OS intrudes into the TS's domain, the relative motion line of OS to the TS will intersect with the boundary of the TS's domain. Therefore, by calculating the existence of an intersection point, can determine whether OS has intruded into the TS's domain.

Similarly, by analyzing whether the relative motion line of the TS to OS intersects with the boundary of OS's domain, can determine whether the TS has intruded into OS's domain. The improved membership function is as follows:

$$k_{DCPA} = \begin{cases} 1, & DCPA < r_1 \| p_1 > 0 \| p_2 > 0 \\ \frac{1}{2} - \frac{1}{2}\sin\left(\frac{180°}{r_2 - r_1}(DCPA - \frac{r_2 + r_1}{2})\right), & r_1 < DCPA < r_2 \\ 0, & DCPA \geq r_2 \end{cases} \quad (16)$$

Here, $p_1$ is the number of intersection points between OS's relative motion line to the TS and the boundary of the TS's domain, $p_2$ is the number of intersection points between the TS's relative motion line to OS and the boundary of OS's domain.

*2) Membership function of TCPA*

$$k_{TCPA} = \begin{cases} 1, & TCPA \leq T_1 \\ \left(\frac{T_2 - TCPA}{T_2 - T_1}\right)^2, & T_1 < TCPA \leq T_2 \\ 0, & TCPA > T_2, DCPA > d_4 \end{cases} \quad (17)$$

$$T_1 = \begin{cases} \frac{\sqrt{d_3^2 - DCPA^2}}{v_R}, & DCPA \leq d_3 \\ \frac{DCPA - d_3}{v_R}, & DCPA > d_3 \end{cases} \quad (18)$$

$$T_2 = \frac{\sqrt{d_4^2 - DCPA^2}}{v_R} \quad (19)$$

Here, $d_3$ represents the latest avoidance distance for the burdened ship, $d_4$ represents the distance over which the ship is capable of taking evasive actions.

*3) Membership function of the distance between two ships (D)*

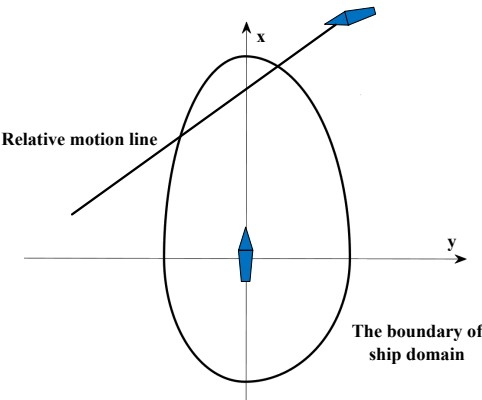

Fig. 3 The relative motion lines of the two ships intersect at the boundary of the ship domain.

$$k_D = \begin{cases} 1, & 0 \leq D \leq d_3 \\ \left(\frac{d_4 - D}{d_4 - d_3}\right)^2, & d_3 < D \leq d_4 \\ 0, & D > d_4 \end{cases} \quad (20)$$

Establish the weight set $W$ based on the importance of each factor in calculating the $CRI$.

$$W = \{W_{DCPA}、\ W_{TCPA}、\ W_D\} \quad (21)$$

where $W_{DCPA} > W_{TCPA} > W_D$ and $W_{DCPA} + W_{TCPA} + W_D = 1$.

$$CRI = W \times k = W_{DCPA}k_{DCPA} + W_{TCPA}k_{TCPA} + W_D k_D \quad (22)$$

*C. Fitness function value (Fitness)*

Incorporate the COLREGs and the CRI into the evaluation of the fitness function, forming an evaluation set F based on factors of safety, economy, compliance with COLREGs, and optimal collision avoidance timing. The CRI determines the safety factor in the fitness function, while the voyage distance and the degree of turning together determine the economic factor in the fitness function.

Constructing CRI-based objective function:

$$F_1 = CRI; \quad (23)$$

In path planning issues, the total voyage determines the consumption of cost during navigation and serves as an important economic assessment index. In the process of navigation, $(x_i, y_i)$ represents the current point, and $(x_{i-1}, y_{i-1})$ is the previous point adjacent to $(x_i, y_i)$, with the total number of path points being n and the destination point being $(x_n, y_n)$. Constructing total voyage-based objective function:

$$F_2 = \frac{\sqrt{(x_i - x_{i-1})^2 + (y_i - y_{i-1})^2}}{\sqrt{(x_n - x_{i-1})^2 + (y_n - y_{i-1})^2}} + \frac{\sqrt{(x_n - x_i)^2 + (y_n - y_i)^2}}{\sqrt{(x_n - x_{i-1})^2 + (y_n - y_{i-1})^2}} \quad (24)$$

Constructing degree of turning-based objective function:

$$F_3 = \arccos\left( \frac{(x_i - x_{i-1}, y_i - y_{i-1}) \cdot (x_n - x_i, y_n - y_i)^T}{\| (x_i - x_{i-1}, y_i - y_{i-1}) \cdot (x_n - x_i, y_n - y_i) \|} \right) \quad (25)$$

Based on the encounter situations and responsibility allocation described earlier, the objective function constructed according to the COLREGs is:

$$F_4 = \begin{cases} 1 & 000° \le \theta_r \le 112.5° \text{ or } 355° \le \theta_r \le 005° \\ 0 & \text{otherwise} \end{cases} \quad (26)$$

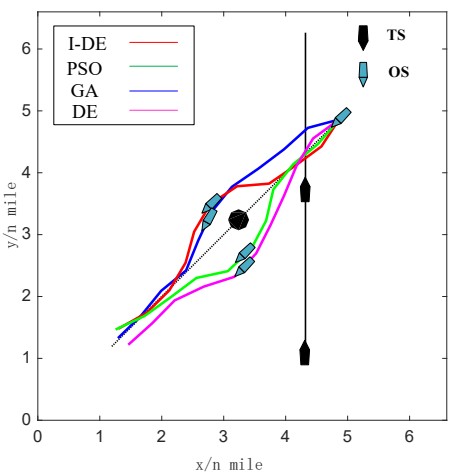

Fig. 4 Simulation results of a two-ship encounter

TABLE II.
SHIP PARAMETERS.

| Parameter | OS | TS |
|---|---|---|
| Length Overall /m | 67.80 | 146.00 |
| Beam/m | 16.00 | 36.00 |
| Draft/m | 2.633 | 3.500 |
| Displacement/t | 1850 | 6530.5 |
| Water density/m³ | 1.025 | 1.025 |

TABLE III.
INITIAL STATES OF THE EXPERIMENTAL OBJECTS IN TWO SHIP ENCOUNTER

| Ship | Initial heading/° | Initial speed/kn | Distance from OS/n mile |
|---|---|---|---|
| OS | 225 | 12 | 0 |
| TS | 0 | 12 | 3.51 |
| Obstacle | none | none | 2.14 |

The timing of collision avoidance depends on the CRI value. Therefore, based on previous research, an objective function for optimal collision avoidance timing is constructed with a threshold value of 0.3:

$$F_5 = \begin{cases} 1 & CRI \ge 0.3 \\ 0 & CRI < 0.3 \end{cases} \quad (27)$$

$$Fitness = W_1 F_1 + W_2 F_2 + W_3 F_3 + W_4 F_4 + W_5 F_5 \quad (28)$$

where $W_1$, $W_2$, $W_3$, $W_4$, $W_5$ are the weights for safety, total voyage, degree of turning, COLREGs, and optimal collision avoidance timing, respectively.

## IV. EXPERIMENT

This paper validates the effectiveness of the proposed Improved Differential Evolution (I-DE) algorithm through simulation experiments on the Matlab platform. The algorithm is compared with the traditional DE, PSO, and GA, using safety and economy as evaluation criteria.

The simulation experiments are set in open waters with good visibility, ignoring external factors such as wind, waves, and currents. The experiments simulated scenarios involving two-ship and three-ship encounters, incorporating static obstacles in the two-ship encounters to more comprehensively assess the autonomous obstacle avoidance capability of the algorithm.

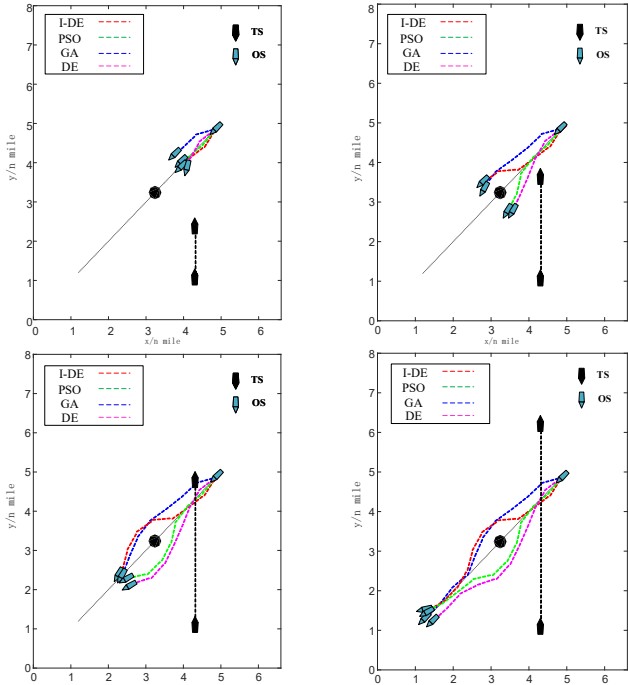

Fig. 5 The states of the ships at specific time intervals in two ship encounter

TABLE IV.
INITIAL STATES OF THE EXPERIMENTAL OBJECTS IN THREE SHIP ENCOUNTER

| Ship | Initial heading/° | Initial speed/kn | Distance from OS/n mile |
|---|---|---|---|
| OS | 50 | 12 | 0 |
| TS1 | 180 | 12 | 6.45 |
| TS2 | 280 | 12 | 6.64 |

TABLE V.
VARIOUS DATA OF SIMULATION RESULTS IN TWO SHIP ENCOUNTER

| Algorithms | I-DE | PSO | GA | DE |
|---|---|---|---|---|
| Min Dis to obstacle/n mile | 0.499917 | 0.404968 | 0.421491 | 0.464535 |
| Min Dis to TS/n mile | 1.241208 | 0.639673 | 1.327054 | 0.561472 |
| Sum deviation Dis/n mile | 8.960443 | 9.832715 | 22.624526 | 10.548143 |
| Runtime/s | 6.7743 | 5.1392 | 12.1593 | 5.407 |

TABLE VI.
VARIOUS DATA OF SIMULATION RESULTS IN THREE SHIP ENCOUNTER

| Algorithms | I-DE | PSO | GA | DE |
|---|---|---|---|---|
| Min Dis to TS1/n mile | 2.409358 | 2.343583 | 2.593572 | 2.426915 |
| Min Dis to TS2/n mile | 1.010833 | 0.916672 | 1.176653 | 1.083573 |
| Sum deviation Dis/n mile | 22.762838 | 47.657674 | 31.220230 | 28.404747 |
| Runtime/s | 10.0148 | 8.352 | 18.726 | 9.7261 |

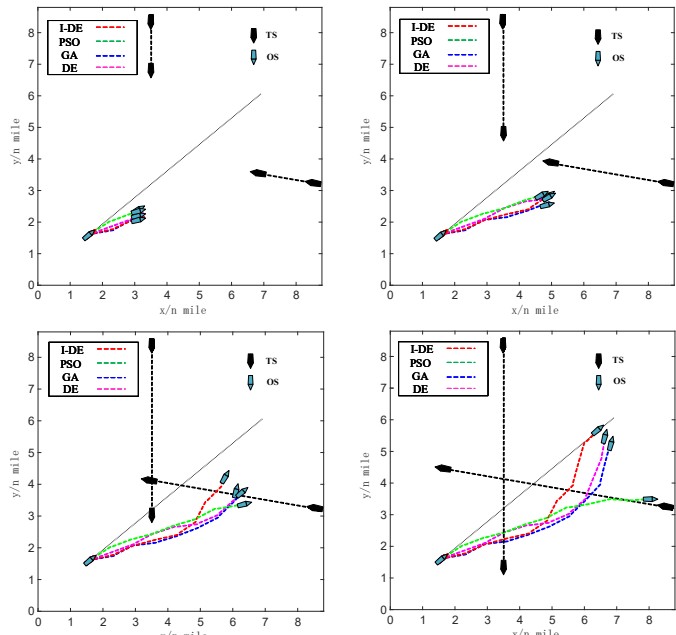

Fig. 7 The states of the ships at specific time intervals in three ship encounter

### A. Two-ship encounters

The outcomes of the simulation are illustrated in Fig. 4-5. Fig. 4 displays the complete paths planned by the various algorithms, while Fig. 5 displays the states of the ships at specific time intervals, reflecting the real-time distance between the OS and the TS, providing data support for evaluating safety. In these figures, the blue ship represents OS, the black ship represents the TS, and the black hexagons represent the static obstacles. The simulation results are detailed in Table IV. By comparing the minimum distances(MD) between the OS and the TS and between the OS and the obstacle, the safety of the paths can be evaluated. The total deviation distance between the planned path and the initial path can be used to assess the economy of the path. Additionally, the table also shows the runtime of each algorithm.

As evidenced by the experimental data, the PSO algorithm and the traditional DE algorithm perform poorly in terms of navigational safety. The paths planned by these two algorithms result in OS maintaining a relatively close distance to the TS during navigation, posing a higher collision risk. In contrast, the paths planned by the I-DE algorithm and the GA maintain a greater distance between the ships, better ensuring navigational safety. By comparing the MD between OS and the obstacle, it is evident that the I-DE algorithm also performs best in avoiding collisions with an obstacle, presenting the lowest collision risk.

As shown in Fig. 4, in the initial stage, there is a considerable distance between the OS and the obstacle, and no evasive action is needed. However, the path planned by the GA deviates from the original route at the outset. This premature maneuver increases the deviation distance of OS, thereby reducing the path's economic efficiency. In contrast, the paths planned by the I-DE algorithm, the PSO algorithm, and the traditional DE algorithm closely adhere to the original path when far from the

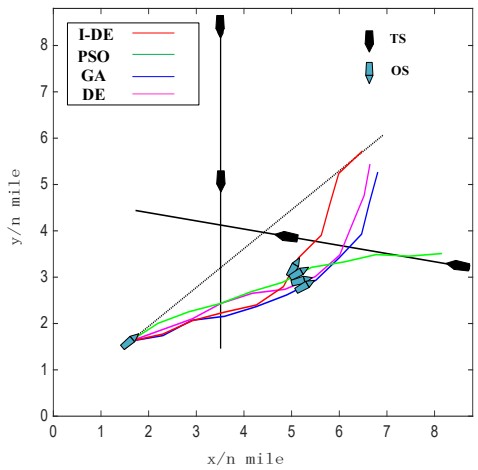

Fig. 6 Simulation results of a three-ship encounter

According to Rule 8 of the COLREGs: In open waters with ample space, changing course is the most effective measure to prevent collisions at close range, provided that timely and effective evasive maneuvers do not result in the two ships coming closer than the safe distance again. Therefore, to facilitate the study, when considering encounter situations and avoidance methods, and following the collision avoidance rules and the usual practices followed by seafarers at sea, the USV, when acting as the give-way ship, avoids a collision by changing course instead of reducing speed or stopping. The parameters and initial states of the experimental objects are displayed in Table II to Table IV.

To ensure that the algorithms are tested under fair conditions, the parameters used in the experiments are uniformly set as follows: total iteration times $K = 50$, initial population size $M = 40$, and dimensions of each individual $n = 5$, $W_{DCPA} = 0.4$, $W_{TCPA} = 0.4$, $W_D = 0.2$, $W_1 = 0.4$, $W_2 = 0.2$, $W_3 = 0.2$, $W_4 = 0.1$, $W_5 = 0.1$.

obstacle. These algorithms start taking evasive actions at approximately 1 nautical mile from the obstacle, adjusting the course to avoid collisions and passing the obstacle from a greater distance. Once the obstacle is safely passed, the ship gradually returns to the original path and eventually reaches the target point. According to the data in Table V, the total deviation distance for the GA is the largest, while the other three algorithms have relatively smaller deviation distances, indicating better economic efficiency.

### B. Three-ship encounters

Similar to the two-ship encounter scenario, the simulation results are presented in Fig. 6 and Fig. 7. The experimental findings indicate that the I-DE algorithm, along with the other three algorithms, ensures that the ship safely navigates through encounters with target ships, successfully avoiding collisions and demonstrating good safety performance.

However, there are significant differences in economic performance among the four algorithms. The PSO algorithm results in the ship deviating significantly from its original course after avoiding TS2, with the total deviation distance reported in Table VI being 47.65 nautical miles, which compromises the economic efficiency of the path. In contrast, the paths planned by the I-DE, DE, and GA algorithms show a tendency to approach the original course after the avoidance maneuver with TS2. As shown in Fig. 6, the I-DE algorithm enables the ship to smoothly return to its original course after avoidance. According to the data in Table VI, the total deviation distances for the I-DE, DE, and GA algorithms are 22.76 nautical miles, 28.40 nautical miles, and 31.22 nautical miles, respectively, indicating that the path planned by the I-DE algorithm exhibits better economic efficiency.

Since the runtime is of lower importance in path evaluation and the differences in runtime among the algorithms are minimal, the impact of runtime is not considered. Considering both safety and economy, the I-DE algorithm demonstrates better performance in planning safe, economical paths, significantly outperforming the other algorithms. Therefore, the simulation experiments confirm the effectiveness of the improved differential evolution algorithm.

## V. Conclusion

To address the issue of path planning and collision avoidance for USVs in open waters, this paper proposes an improved adaptive differential evolution algorithm. This algorithm adaptively adjusts the crossover factor (CR) in the crossover operation, enhancing population diversity while increasing the independence among individuals. Additionally, the CRI is incorporated into the fitness function evaluation. A restriction factor is added when selecting the membership function of DCPA, making the calculation of the collision risk more consistent with maritime practices. Simulations were conducted on the Matlab platform, comparing the I-DE algorithm with the PSO algorithm, the GA, and the traditional DE algorithm under two-ship encounter and three-ship encounter scenarios. Safety is determined by the minimum distance between own ship and the obstacle and the target ship, and economy is determined by the total deviation distance. The experimental outcomes indicate

that the I-DE algorithm outperforms others in terms of safety and economy, thereby validating its effectiveness.

In future research on path planning and collision avoidance for USVs, it is essential to fully consider the interference of external factors as well as the maneuverability of the USV. Furthermore, attention should be given to aspects such as communication delays, human factors in decision-making, and integration with existing maritime traffic management systems, thereby ensuring closer alignment with maritime practices.

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
