# OpenReview forum: "Path Planning of USV Based on the Improved Differential Evolution Algorithm"
_IEEE.org/ICIST/2024/Conference — IEEE ICIST 2024 Conference Submission_

### Official Review · Reviewer_GN7M · 2024-08-24
**This article is well written, but there are still some minor errors.**

**Rating:** 7
**Confidence:** 4

**Review:**

The 'where' in the text should be lowercase and cannot be indented before it. Please read the entire text carefully and correct this error.

---

### Official Review · Reviewer_FhTZ · 2024-08-29
**Accept**

**Rating:** 10
**Confidence:** 5

**Review:**

This paper focuses on introducing an improved adaptive differential evolution algorithm. Overall, the paper presents a novel study that introducing this algorithm can effectively solve the issue of path planning and collision avoidance for USVs in open waters. However, several aspects could benefit from clarification, revision, or additional analysis.
1. Some figures could be improved by reorganizing data or adding labels.
2. The author should consider additional comparative experiments or simulations to strengthen conclusions.
3. The discussion of limitations and future work could be expanded to more fields.

---

### Official Review · Reviewer_pz14 · 2024-09-03
**comment**

**Rating:** 7
**Confidence:** 4

**Review:**

This paper introduces a differential evolution algorithm (DE) with an adaptive crossover factor for path planning and collision avoidance in USV. The collision risk index (CRI) is integrated with the DE, and the CRI is improved by introducing a restriction factor when selecting the degree of membership for the distance to closest point of approach (DCPA). The experimental results demonstrate that, compared with the other three algorithms, the improved DE exhibits greater advantages in terms of minimum distance to the target ship, minimum distance to obstacles, and total yaw distance. In reviewer's opinion, this paper is acceptable.

---

### Decision · Program_Chairs · 2024-09-06

Accept (Oral)